# TRIM28 is a target for paramyxovirus V proteins

Gauthier Lieber[1], Florence Kwaschik[1], Marie Lork[1], Nora Schmidt[2], Benjamin G. Hale [1,*]

**1** Institute of Medical Virology, University of Zurich, Zurich, Switzerland, **2** Institute of Virology, Hannover Medical School, Hannover, Germany

\* hale.ben@virology.uzh.ch

## Abstract

SUMO-modified Tripartite Motif Protein 28 (TRIM28; KAP1) plays a crucial role in repressing endogenous retroelement (ERE) transcription. We previously provided evidence that loss of SUMO-modified TRIM28 triggered by influenza A virus (IAV) infection promotes activation of host antiviral immunity via a mechanism involving derepression of EREs and production of immunostimulatory RNAs. While the IAV NS1 protein might limit consequences of such activation via its dsRNA-binding activity, we hypothesized that other human pathogenic viruses could have evolved more direct strategies to counteract this potential ERE-based defense system. Here, we reveal that V proteins from diverse paramyxoviruses, including Measles, Mumps, Parainfluenza, and Nipah/Hendra viruses, can all engage with TRIM28. Notably, the efficiency of engagement varies markedly between virus species, a phenotype that can be linked to specific residues within the C-terminal domain of V proteins. Further mapping showed that V proteins target both the TRIM28 Coiled-Coil domain and the TRIM28 PHD-Bromodomain, which contains the functionally-relevant TRIM28 SUMO-modification sites necessary for ERE repression. In this context, while paramyxovirus infection triggers canonical stress-associated phosphorylation of TRIM28, loss of SUMO-modified TRIM28 does not occur, and minimal induction of the TRIM28-dependent ERE RNA, HERVK14C, is observed. Furthermore, pre-infection with Parainfluenza virus type 2, which encodes a V protein that efficiently engages with TRIM28, limits subsequent IAV-triggered loss of SUMO-modified TRIM28 and upregulation of HERVK14C RNA. These findings dissect the interplay between paramyxoviruses and TRIM28, providing support for the concept of a TRIM28-regulated ERE-based antiviral defense system by uncovering a potential viral antagonistic measure.

## Author summary

Host recognition of viral nucleic acids is fundamental for stimulating antiviral immunity. To prevent aberrant immune activation, cells must precisely differentiate

**PLOS Pathogens**

**Data availability statement:** All relevant data are within the manuscript and its Supporting information files.

**Funding:** The research leading to these results received funding from the Swiss National Science Foundation (www.snf.ch; grant 310030_214957 to BGH) and the Hartmann Müller Foundation (www.hms.uzh.ch; grant 2378 to BGH). The funders had no role in study design, data collection, data interpretation, or the decision to submit the work for publication.

**Competing interests:** The authors have declared that no competing interests exist.

"non-self" from "self" nucleic acids. Endogenous retroelements (EREs), despite their self-origin, can produce immunostimulatory RNAs that blur this distinction, and may have been functionally co-opted to enhance antiviral immunity in response to some infections. Normally, ERE transcription is repressed by the host protein TRIM28, the function of which is dynamically modulated by SUMO-modification. Here, we demonstrate that paramyxoviruses engage TRIM28, maintaining it in its repressive, SUMO-modified form during infection. We propose that this virus-host interaction limits ERE activation and consequently suppresses ERE-driven immune signaling, revealing a previously unidentified mechanism of immune evasion by paramyxoviruses.

## Introduction

Endogenous retroelements (EREs) constitute approximately 45% of the human genome, and comprise various classes of mobile genetic elements, including long interspersed nuclear elements (LINEs), short interspersed nuclear elements (SINEs), and endogenous retroviruses (ERVs) [1]. While DNA transposons have the potential to move within the genome via a "cut-and-paste" mechanism, EREs utilize a "copy-and-paste" mechanism that involves RNA intermediates [2,3]. EREs such as ERVs, which originate from ancient retroviruses that have integrated into the germline, are typically flanked by one or two Long Terminal Repeats (LTRs) that facilitate their own RNA transcription and can also act as promoters regulating expression of nearby genes [2–4]. Thus, activity and expression of EREs can disrupt cellular homeostasis in various ways, not only by potential interference with genome integrity via insertional mutagenesis, but also by splicing modulation, epigenetic alterations, and promoter/enhancer activity [3,5,6]. Additionally, the transcription of some EREs has been proposed to produce double-stranded RNA (dsRNA) as a consequence of bidirectional transcription from ERV LTRs or via read-through transcription and complementarity found in some ERV sequence arrangements [7]. Such ERE-dsRNA, while technically self-derived, can behave as a mimetic of exogenous virus-derived dsRNA, and thereby trigger unwarranted immune or pro-inflammatory responses by activating antiviral pathways [6,8–10]. Aberrant expression of EREs has therefore been linked to some autoimmune pathologies, such as inflammatory bowel disease, where elevated transcript levels of EREs, including HERV-H and HERV-K, have been observed in affected individuals [11]. To mitigate these potential deleterious effects, strict repression of ERE expression is critical. Epigenetic silencing mechanisms play a central role in this regulation, and typically include DNA methylation and histone modifications. A key pathway mediating such silencing mechanisms involves Krüppel-associated box (KRAB) zinc-finger proteins (ZFPs), which bind specific DNA sequences within ERE loci and, in concert with SUMO-modified Tripartite Motif Protein 28 (TRIM28; KAP1), recruit the methyltransferase SETDB1 to enforce transcriptional repression [12–14].

In contrast to being entirely negative in consequence, recent studies have generated interest in the concept that ERE activity may have an important functional role,

in particular with respect to host immune defenses [6,10,15,16]. For example, some ERE regulatory elements have been shown to function as enhancers driving the expression of specific innate immune genes in response to antiviral signals [5,6]. Moreover, increasing evidence suggests that the pathogen-associated molecular pattern (PAMP)-like properties of some transcripts generated from EREs might be used by the host to further potentiate interferon (IFN) responses during exogenous virus infections [6]. In this regard, we previously provided evidence for a host antiviral mechanism triggered by influenza A virus (IAV) infection that involves loss of SUMO-modified TRIM28 and the consequent upregulation of ERE transcripts in infected cells [17], some of which appear to form a source of cellular ERE-dsRNA that may be sensed by various antiviral pattern-recognition receptors (PRRs) [18]. Notably, this phenomenon may not be unique to IAV infections, as other viruses (such as SARS-CoV-2, HCV, and HSV-1) have also been reported to induce high expression levels of EREs during infection [19–21]. This suggests that infection-triggered ERE derepression may be a co-opted response as part of a broader immune pathway, leading us to hypothesize that some viruses could have evolved antagonistic strategies to counteract this defense mechanism at different levels. For example, we reported that the IAV non-structural protein 1 (NS1) has the capacity to interact with infection-induced ERE-dsRNA and potentially to limit ERE-dsRNA engagement with host antiviral dsRNA sensors in the cytosol, thus likely dampening immune activation via this pathway [18]. Whether additional viruses have evolved similar or unique strategies to limit ERE derepression or sensing is currently unknown. However, it is notable that proteins from several DNA viruses, such as adenovirus E1B-55K [22], Kaposi's sarcoma-associated herpesvirus vPK [23]/LANA [24,25], and AAV Rep52 [26], can interact with TRIM28, albeit to release viral DNA genomes from direct TRIM28-mediated transcriptional repression [23,27]. We therefore hypothesized that other types of viruses also express proteins that engage with TRIM28, but which might limit its function in infection-triggered ERE derepression and potentiation of immune activation.

In this study, we mined an existing large-scale proteomic dataset to uncover RNA virus proteins that may interact with TRIM28 [28]. We limited our search to RNA viruses in order to exclude the identification of TRIM28 interactors that likely aid viral DNA genome derepression. Our survey revealed two potential TRIM28 interactor candidates: Measles virus (MeV; *Paramyxoviridae*) V and La Crosse virus (*Bunyaviridae*) NSs [28]. Here, we focused on V protein, a potent virulence factor expressed by many *Paramyxoviridae* family members that antagonizes the IFN system at multiple independent levels [29–32]. Using co-immunoprecipitation assays, we show that the ability to engage with TRIM28 is a broadly conserved property of V proteins from across the *Paramyxoviridae* family, albeit interaction efficiencies vary between viruses. Importantly, the C-terminal domain of V is necessary and sufficient for TRIM28 engagement, and domain-mapping revealed that one target for V is the region of TRIM28 that harbors the functionally-relevant TRIM28 SUMO-modification sites. Notably, both paramyxovirus and IAV infections were observed to trigger phosphorylation of TRIM28 at S824, which is associated with chromatin relaxation [33,34], but only IAV infection caused a loss of SUMO-modified TRIM28 and upregulation of TRIM28-dependent ERE transcripts. Furthermore, pre-infection of cells with a paramyxovirus expressing a V protein that efficiently engages TRIM28 was able to partially inhibit subsequent IAV-induced loss of SUMO-modified TRIM28, as well as limit IAV-induced upregulation of a TRIM28-dependent ERE transcript. This study therefore provides further insights into the interplay between RNA viruses and TRIM28. The identification of paramyxovirus V protein engagement with TRIM28 supports the concept that some viruses may have evolved countermeasures to TRIM28-regulated ERE-based antiviral defense systems.

## Results

### Paramyxovirus V proteins engage with TRIM28

In an effort to identify potential viral antagonists of TRIM28-regulated ERE-based antiviral defense systems, and thus provide evidence for its biological significance, we surveyed an existing proteomic dataset of virus-host protein-protein interactions to uncover viral proteins that could co-immunoprecipitate human TRIM28 [28]. From 70 viral proteins tested, the V protein of MeV and the NSs protein of La Crosse virus were identified as candidate factors potentially able to interact with TRIM28 [28].

We initially focused on the V protein and proceeded to verify this interaction through targeted co-immunoprecipitation (co-IP) assays, while also expanding the initial observation to a wider set of diverse paramyxovirus V proteins. Thus, HEK293T cells were transfected with plasmids expressing FLAG-tagged mCherry or FLAG-tagged V proteins from Hendra virus (HeV), Mapuera virus (MapV), MeV, Menangle virus (MenV), Mumps virus (MuV), Newcastle Disease virus (NDV), Nipah virus (NiV), Parainfluenza virus 2 (PIV2), Parainfluenza virus 5 (PIV5), Sendai virus (SeV), or Tioman virus (TioV). These paramyxoviruses were selected as key representatives of the *Orthorubulavirus* (PIV2, PIV5, MuV, MapV), *Pararubulavirus* (TioV, MenV), *Respirovirus* (SeV), *Henipavirus* (HeV, NiV), *Orthoavulavirus* (NDV), and *Morbillivirus* (MeV) genera within the *Paramyxoviridae* family. 24 hours post-transfection, total soluble cell lysates were prepared and used to immunoprecipitate the FLAG-tagged proteins together with co-precipitating proteins. Subsequent western blot analysis revealed that TRIM28 could be specifically co-precipitated with all FLAG-tagged V proteins tested, but not with the FLAG-tagged mCherry control (Fig 1A). While this result indicates a broadly conserved engagement of paramyxovirus V proteins and TRIM28, quantification of western blot band intensities revealed clear differences in the relative efficiencies of TRIM28 co-precipitation by each V protein (Fig 1B). For example, PIV2-V consistently co-precipitated the most TRIM28, while MeV-V demonstrated mid-level efficiency, and PIV5-V and NiV-V exhibited amongst the poorest efficiencies (Fig 1B). To validate the interaction observation under infection conditions, we infected the human lung carcinoma cell-line, A549, with PIV5 at a multiplicity of infection (MOI) of 2 PFU/cell, and 24 hours later prepared total soluble cell lysates for co-IP studies using an anti-TRIM28 antibody and an anti-V5 antibody, which targets a common epitope on both the PIV5 V and P proteins. Consistent with the results from our transfection-based assays, we observed co-precipitation of TRIM28 with anti-V5 antibody only from infected cell lysates (Fig 1C). Reciprocally, V/P proteins were specifically co-precipitated with the anti-TRIM28, but not control, antibody from PIV5-infected cells (Fig 1D). Importantly, P was not co-precipitated with TRIM28 from cells infected with a recombinant PIV5 lacking a functional V protein (PIV5 VΔC) [35], indicating that TRIM28 is specifically targeted by V, and not P (Fig 1D). While several known functions of V are thought to occur in the cytoplasm [29–31], there is extensive infection- and transfection- based evidence to indicate that many V proteins can be found in both the cytoplasm and nucleus [36–43], where TRIM28 primarily resides. Indeed, immunofluorescence microscopy of HEK293T cells transfected with a panel of plasmids expressing FLAG-tagged V proteins confirmed simultaneous cytoplasmic and nuclear localizations for most paramyxovirus V proteins tested (Fig 1E). This observation highlights that V proteins and TRIM28 have the capacity to engage physically with one another in the nucleus. Interestingly, NiV-V protein is strongly cytoplasmic under steady-state conditions (Fig 1E) [37], and it is tempting to speculate that its limited nuclear trafficking may contribute to inefficient targeting of TRIM28 as compared with other paramyxovirus V proteins (Fig 1B).

## Residues in the C-terminal domain of V proteins determine engagement with TRIM28

We next sought to identify regions of V proteins critical for the co-precipitation of TRIM28. To this end, HEK293T cells were transfected with plasmids expressing FLAG-tagged full-length (FL) versions of V proteins from MeV, MuV, or PIV5, alongside FLAG-tagged truncated versions of each V protein comprising only their C-terminal domain (CTD) or N-terminal domain (NTD). Parallel transfection of HEK293T cells with a plasmid expressing FLAG-tagged mCherry acted as a negative control. 24 hours post-transfection, total soluble cell lysates were prepared and used to immunoprecipitate the FLAG-tagged proteins together with any co-precipitating proteins. Subsequent western blot analysis revealed that, as expected, TRIM28 could be specifically co-precipitated with all the FLAG-tagged FL V proteins tested, but not with the FLAG-tagged mCherry control (Fig 2A). Furthermore, TRIM28 could be readily co-precipitated with each V protein CTD (despite their very low expression), but not with any V protein NTD, indicating that paramyxovirus V protein CTDs are necessary and sufficient for engagement with TRIM28 (Fig 2A). As paramyxovirus P/V proteins can share the same NTD, these results further support the notion that TRIM28 is a specific target of V, and not P.

Given our observation that PIV2-V protein co-precipitated the most amount of TRIM28 as compared with other paramyxovirus V proteins (Fig 1B), we next sought to understand the molecular basis for this phenotype. Notably, among

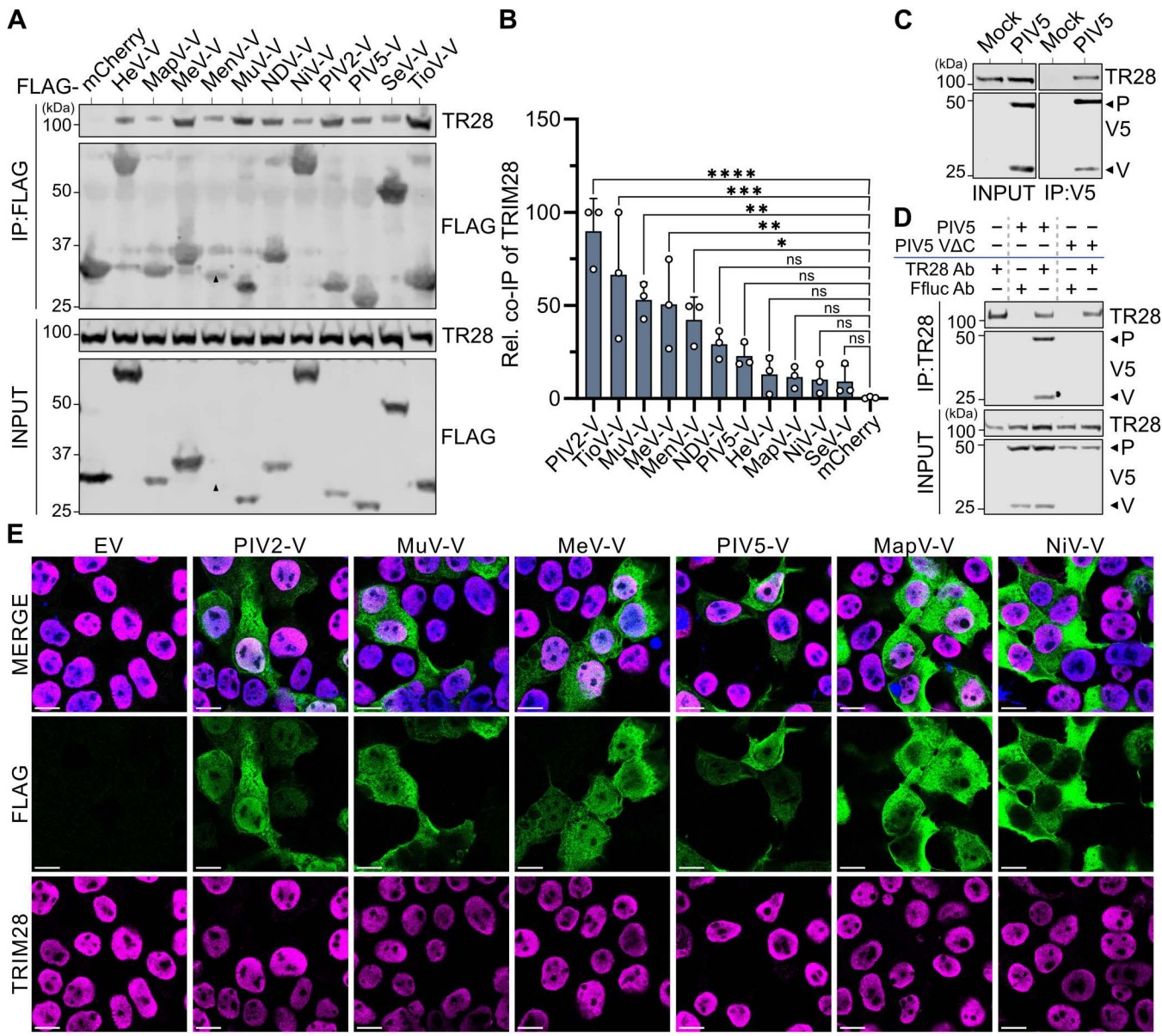

**Fig 1. Paramyxovirus V proteins engage with TRIM28.** (A) V proteins from 11 different paramyxoviruses can co-precipitate TRIM28. HEK293T cells were transfected with plasmids expressing the indicated FLAG-tagged V constructs or FLAG-tagged mCherry for 24 hours. FLAG-tagged proteins were subsequently immunoprecipitated from cell lysates, and the indicated proteins were detected by western blotting. IP: immunoprecipitated fraction; INPUT: soluble input fraction; TR28: TRIM28. Data are representative of n = 3 independent experiments. Black arrows indicate predicted sizes of selected proteins of interest. (B) Quantification shows TRIM28 co-precipitation levels differ between V proteins. Band intensities of co-precipitated TRIM28 from experiments shown in (A) were normalized to their input levels and expressed relative (rel.) to the levels of the respective immunoprecipitated FLAG-tagged protein. Data points correspond to results from individual experiments, and results from n = 3 independent experiments are shown. Bars represent mean values, with error bars representing standard deviation (SD). Statistical significance was assessed by ordinary one-way ANOVA (*, P ≤ 0.05; **, P ≤ 0.01; ***, P ≤ 0.001; ****, P ≤ 0.0001; ns, non-significant). (C) and (D) V and TRIM28 proteins can be co-precipitated from infected cells. A549 cells were infected, or mock, with PIV5 or PIV5-VΔC (MOI = 2 PFU/cell) for 24 hours, and soluble cell lysates were prepared. (C) Immunoprecipitations were performed using an anti-V5 antibody, and the indicated co-precipitated proteins were analyzed by western blot. (D) Immunoprecipitations were performed using an anti-TRIM28 antibody (TR28) or an anti-Ffluc antibody (control), and the indicated co-precipitated proteins were analyzed by western blot. Input fractions were also analyzed. Data are representative of n = 3 independent experiments. (E) Immunofluorescence microscopy reveals V proteins can exhibit cytoplasmic and nuclear localization patterns. HEK293T cells were transfected with plasmids expressing the indicated FLAG-tagged V constructs or FLAG-tagged empty vector (EV) for 24 hours. Cells were subsequently fixed, permeabilized, and stained with DAPI (nuclei, blue), anti-FLAG antibody (V proteins, green), and anti-TRIM28 antibody (TRIM28, magenta). Images were acquired by confocal microscopy. Scale bars: 10 μm. Data are representative of n = 2 independent experiments.

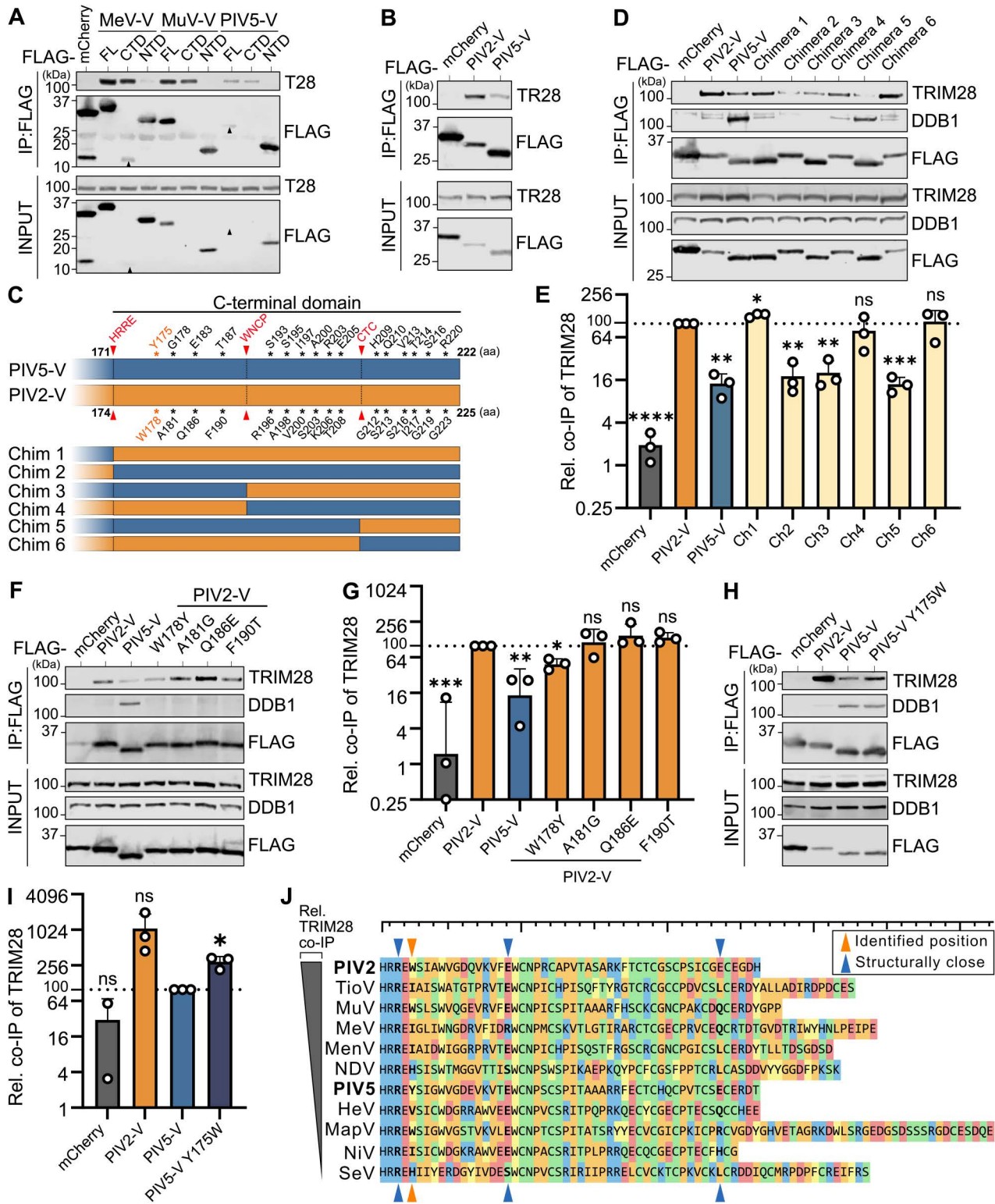

**Fig 2. Residues in the C-terminal domain of V proteins determine engagement with TRIM28.** (A) The C-terminal domain of V proteins is necessary and sufficient for TRIM28 engagement. HEK293T cells were transfected for 24 hours with FLAG-tagged constructs expressing mCherry or the MeV-V,

MuV-V, or PIV5-V proteins in their full-length (FL), C-terminal domain (CTD), or N-terminal domain (NTD) forms. FLAG-tagged proteins were subsequently immunoprecipitated and the indicated co-precipitated proteins analyzed by western blot (IP: FLAG). Input fractions were also analyzed. Data are representative of n = 3 independent experiments. Black arrows indicate predicted sizes of selected proteins of interest. (B) Comparison of TRIM28 co-precipitation abilities between PIV2-V and PIV5-V. HEK293T cells were transfected for 24 hours with plasmids expressing the indicated FLAG-tagged constructs prior to anti-FLAG immunoprecipitations and analyses as in (A). Data are representative of n = 3 independent experiments. (C) Schematic representation of CTD differences between PIV2-V and PIV5-V. Conserved motifs, serving as cut-off points for chimeras, are highlighted in red, and differential amino acids (aa) are noted. Chimeric constructs (Chim 1–6) were designed by swapping segments between PIV2-V (orange) and PIV5-V (blue) as indicated. (D) Comparison of TRIM28 and DDB1 co-precipitation abilities between chimeric V constructs. HEK293T cells were transfected for 24 hours with plasmids expressing the indicated FLAG-tagged constructs and chimeras depicted in (C) prior to anti-FLAG immunoprecipitations and analyses as in (A). Data are representative of n = 3 independent experiments. (E) Quantification of TRIM28 co-precipitation by chimeric V constructs. Band intensities of co-precipitated TRIM28 from experiments shown in (D) were normalized to their input levels and expressed relative (rel.) to the levels precipitated by PIV2-V. Data points correspond to results from individual experiments, and results from n = 3 independent experiments are shown. Bars represent mean values, with error bars representing standard deviation (SD). Statistical significance was assessed using a one-sample t test (*, $P \leq 0.05$; **, $P \leq 0.01$; ***, $P \leq 0.001$; ****, $P \leq 0.0001$; ns, non-significant). (F) Comparison of TRIM28 and DDB1 co-precipitation abilities between PIV2-V mutants. HEK293T cells were transfected for 24 hours with plasmids expressing the indicated FLAG-tagged constructs and mutants prior to anti-FLAG immunoprecipitations and analyses as in (A). Data are representative of n = 3 independent experiments. (G) Quantification of TRIM28 co-precipitation by PIV2-V mutants. Band intensities of co-precipitated TRIM28 from experiments shown in (F) were normalized to their input levels and expressed relative (rel.) to the levels precipitated by PIV2-V. Data points correspond to results from individual experiments, and results from n = 3 independent experiments are shown. Bars represent mean values, with error bars representing standard deviation (SD). Statistical significance was assessed using a one-sample t test (*, $P \leq 0.05$; **, $P \leq 0.01$; ***, $P \leq 0.001$; ns, non-significant). (H) Comparison of TRIM28 and DDB1 co-precipitation abilities between PIV5-V and PIV5-V Y175W. HEK293T cells were transfected with plasmids expressing the indicated FLAG-tagged constructs prior to anti-FLAG immunoprecipitations and analyses as in (A). Data are representative of n = 3 independent experiments. (I) Quantification of TRIM28 co-precipitation by PIV5-V and PIV5-V Y175W. Band intensities of co-precipitated TRIM28 from experiments shown in (H) were normalized to their input levels and expressed relative (rel.) to the levels precipitated by PIV5-V. Data points correspond to results from individual experiments, and results from n = 3 independent experiments are shown. Bars represent mean values, with error bars representing standard deviation (SD). Statistical significance was assessed using a one-sample t test (*, $P \leq 0.05$; ns, non-significant). (J) Sequence alignment of the V protein CTDs from different paramyxoviruses. The sequences are ordered by their TRIM28 co-precipitation efficiency (top: most efficient; bottom: least efficient). The residue identified as critical for differential PIV2-V/PIV5-V TRIM28 engagement (W178/Y175) is indicated with an orange arrow. Three additional residues in close structural proximity to this residue, based on the PIV5-V structure [73] are indicated with blue arrows.

*Orthorubulaviruses*, the PIV2-V CTD and the PIV5-V CTD are highly similar, with each comprising 53 residues with 80.4% sequence identity, and only differing at 16 amino acid residues. Using side-by-side transfection-based expression and immunoprecipitation assays, we reconfirmed that PIV2-V co-precipitates TRIM28 more robustly than PIV5-V (Fig 2B). We then generated six chimeric FLAG-tagged V protein constructs by swapping CTD segments between PIV2-V and PIV5-V, which were defined by highly-conserved CTD amino acid sequence motifs: HRRE, WCNP, and CTC (Fig 2C). We subsequently assessed the abilities of these FLAG-tagged chimeric constructs to co-immunoprecipitate TRIM28 following transfection into HEK293T cells. As shown in Fig 2D, chimeric constructs 1, 4, and 6 co-precipitated TRIM28 to an extent similar to that of FL PIV2-V, while chimeric constructs 2, 3, and 5 co-precipitated TRIM28 similarly to FL PIV5-V, phenotypes that were highly reproducible over multiple replicates (Fig 2E). This distinct pattern revealed the critical role of the proximal part of the V CTD (sequence between the HRRE and WNCP motifs) in determining engagement with TRIM28. Intriguingly, this region of PIV2-V has previously been implicated in its strong nuclear targeting and retention [36]. Importantly, the differential ability of V protein constructs to engage with TRIM28 was disconnected from the interaction of V with DDB1 (Fig 2D), a key host factor targeted strongly by PIV5-V but less efficiently by PIV2-V [29,44].

The proximal part of the V CTD (between the HRRE and WNCP motifs) exhibits four residue differences between PIV2 and PIV5 (Fig 2C), suggesting that one or more of these residues is responsible for the differential engagement with TRIM28. We therefore introduced individual substitutions into PIV2-V that correspond to the respective PIV5-V residues and used the transfection-based expression and immunoprecipitation assay to assess the potential loss-of-function phenotype in each PIV2-V mutant (Fig 2F). Quantification of independent replicate experiments revealed that only the W178Y substitution in PIV2-V led to a significant reduction in co-precipitated TRIM28 as compared to wild-type PIV2-V, suggesting a critical role for this specific V residue in mediating engagement with TRIM28 (Fig 2G). To further validate this, we generated the reciprocal substitution in PIV5-V (Y175W) and could show that this change conferred an enhanced

ability of PIV5-V to co-precipitate TRIM28 (Fig 2H, I). Overall, these data reveal that the V CTD is the primary mediator of engagement with TRIM28, and that residue differences in this domain can account for differential engagement efficiencies between PIV2 and PIV5.

Sequence alignment of V protein CTDs from the paramyxoviruses tested, ordered by their TRIM28 co-IP ability, did not reveal a clear correlation between the identity of residue 178/175 (PIV2/PIV5 numbering) and TRIM28 engagement efficiency (Fig 2J). Similarly, an assessment of three structurally adjacent residues showed no obvious pattern explaining the differential TRIM28 engagement observed experimentally. Thus, while PIV2/PIV5 V protein residues 178/175 contribute to TRIM28 engagement efficiency, a more complex motif likely determines efficiency in the context of other paramyxovirus V proteins, which could be related to unknown nuclear targeting and/or retention sequences if intracellular localization is a factor in the interaction capacity.

## The TRIM28 PHD-Bromodomain and Coiled-Coil domain are targets for V

We next sought to identify the specific regions of TRIM28 targeted by V. TRIM28 consists of six well-characterized domains: the Really Interesting New Gene domain (RING; harboring E3 ubiquitin ligase activity), two BBox domains (enabling TRIM28 oligomerization), the Coiled-Coil domain (CC; that recruits KRAB-ZNF co-factors), a Linker domain (LNK; containing the HP1 binding site), and the PHD-Bromodomain (PB; essential for self-SUMOylation, and for recruiting the SUMO E2 enzyme, NuRD, and SETDB1) [45]. We therefore generated a series of HA-tagged TRIM28 truncation mutants representing individual or combined TRIM28 domains (Fig 3A). Subsequently, these constructs (alongside HA-tagged FL TRIM28) were transfected into TRIM28-knockout HEK293T cells (HEK293T-T28KO) together with FLAG-tagged mCherry or FLAG-tagged MeV-V, and their ability to be co-precipitated with each FLAG-tagged protein was assessed. MeV-V was chosen for these assays due to its intermediate TRIM28 engagement phenotype (Fig 1B). Strikingly, only HA-tagged TRIM28 constructs containing the CC or PB domains were specifically co-precipitated with FLAG-tagged V protein, whereas constructs lacking both of these domains showed no clear interaction (Fig 3B,C). It was notable that the PB domain alone, which contains all the functionally-relevant TRIM28 SUMO-modification sites required for transcriptional repression, was co-precipitated with V to an extent similar to that of FL TRIM28, which may indicate that PB is the major target of V. Importantly, this domain also contains the phosphorylation site S824, which has been linked to chromatin relaxation [33]. However, MeV-V still efficiently co-precipitated a construct lacking the region containing this site (PBdS) (Fig 3C), indicating that the V-TRIM28 interaction is independent of this phosphorylation event.

Protein sequence alignment of TRIM28 PB and CC domains across multiple potential paramyxovirus host species, including human (e.g., PIV2, PIV5, MeV), bat (e.g., NiV, HeV), horse (NiV, HeV), dog (PIV5), mouse (SeV), and chicken (NDV), revealed a high degree of conservation (Figs 3D and S1). The PB domain is highly conserved among mammalian species, with >90% sequence identity and similarity relative to human TRIM28, whereas chicken TRIM28 PB exhibits much lower conservation (55.9% identity, 71% similarity). Similarly, the CC domain shows near-complete conservation across mammals (99.4% identity, 100% similarity), with chicken TRIM28 again displaying higher divergence (82% identity, 93.6% similarity). Given that NiV and HeV are bat-borne and can infect horses, the high similarity between bat, horse and human TRIM28 suggests that the inefficient engagement of NiV and HeV V proteins with human TRIM28 is not due to a species-specific adaptation.

TRIM28 belongs to the transcriptional intermediary factor 1 (TIF-1) family, a sub-group of TRIM proteins involved in chromatin regulation, transcriptional repression, and genome stability [46]. Alongside TRIM28, all other TIF-1 family members (TRIM24, TRIM33, and TRIM66) also contain PB domains. To determine whether the engagement of V protein with TRIM28 PB extends to other TIF-1 family members, we co-transfected HEK293T-T28KO cells with plasmids expressing FLAG-tagged MeV-V or mCherry, and HA-tagged PB domains of TRIM28, TRIM24, TRIM33, or TRIM66. Strikingly, all TIF-1 PB domains were specifically co-immunoprecipitated by FLAG-tagged MeV-V, indicating that V might have the capacity to engage, directly or indirectly, with multiple members of the TIF-1 family via their PB domains (Fig 3E), although

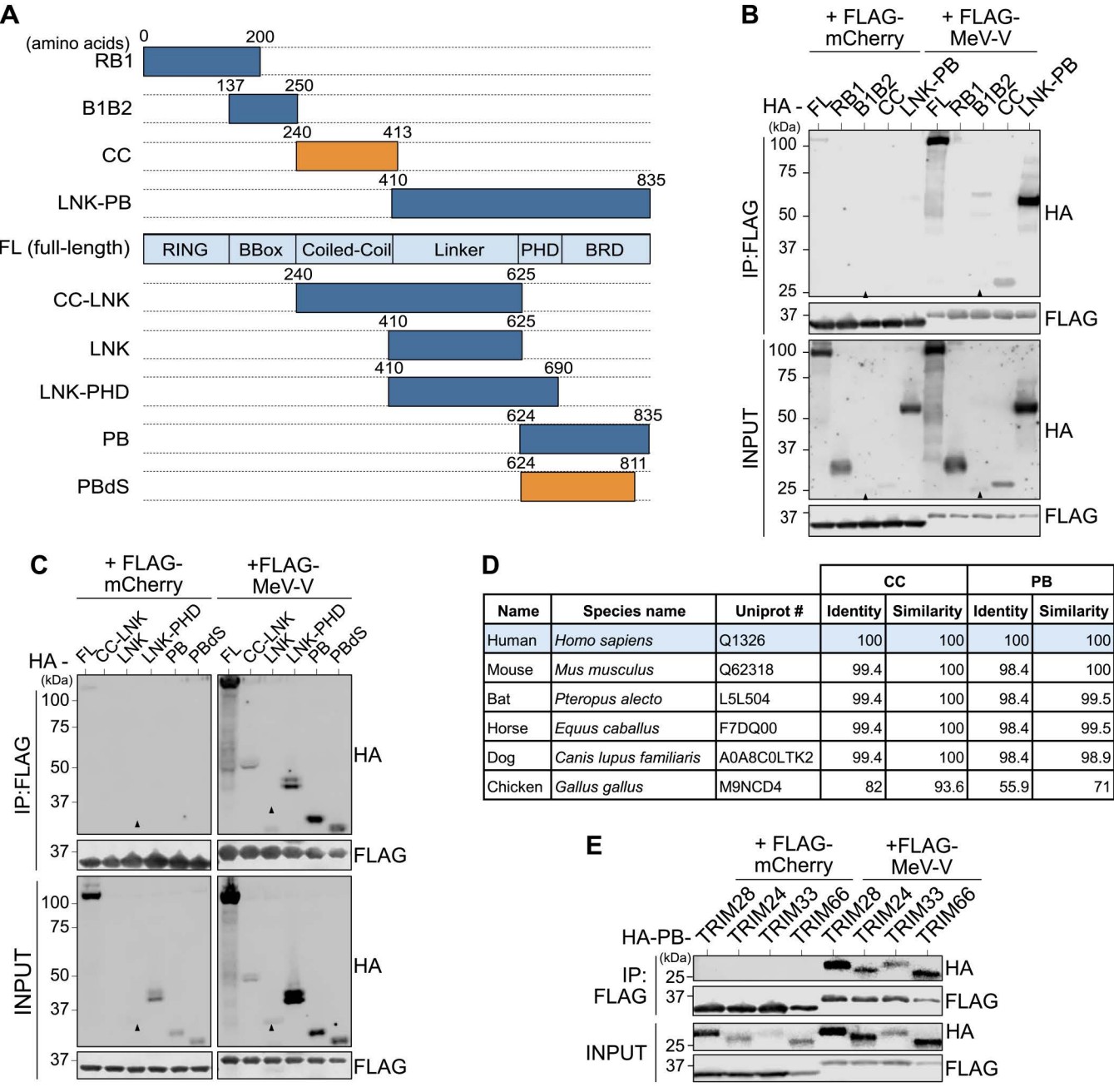

**Fig 3. The TRIM28 PHD-Bromodomain and Coiled-Coil domain are targets for V.** (A) Schematic representation of TRIM28 domains, as well as the respective constructs subsequently used. Numbers indicate full-length (FL) TRIM28 amino acid positions. Constructs retain various combinations of functional domains: Really Interesting New Gene (RING); two BBoxes (B1 and B2); Coiled-Coil (CC); Linker (LNK); PHD; and Bromodomain (BRD). PBdS refers to a construct lacking residues surrounding the S824 phosphorylation site. The smallest domains identified as important for V co-precipitation in subsequent experiments are colored orange. (B) and (C) TRIM28 domains targeted by V. HEK293T-T28KO cells were co-transfected for 24 hours with plasmids expressing FLAG-tagged mCherry or FLAG-tagged MeV-V together with the indicated HA-tagged TRIM28 construct described in (A). FLAG-tagged proteins were subsequently immunoprecipitated and the co-precipitated HA-tagged proteins were analyzed by western blot. Input fractions were also analyzed. Data are representative of n = 3 independent experiments. Black arrows indicate predicted sizes of selected proteins of interest. (D) Sequence conservation in key domains from TRIM28 orthologs among selected potential paramyxovirus host species. Percentage amino acid identity and similarity scores between the PHD-Bromodomain (PB) or Coiled-Coil (CC) domain of the indicated TRIM28 proteins were determined using Clustal Omega. (E) TIF-1 family member PB domains are also potential targets for V proteins. HEK293T-T28KO cells were co-transfected for 24 hours with plasmids expressing FLAG-tagged mCherry or FLAG-tagged MeV-V together with the indicated HA-tagged PB domains from TRIM28,

TRIM24, TRIM33, or TRIM66. FLAG-tagged proteins were subsequently immunoprecipitated and the co-precipitated HA-tagged proteins were analyzed by western blot (IP: FLAG). Input fractions were also analyzed. Data are representative of n = 2 independent experiments.

we cannot dissect contributions from the endogenous full-length TIF-1 family member proteins in this experimental system. Notably, while the function of TRIM66 is poorly characterized, TRIM24, TRIM28, and TRIM33 all reportedly form complexes that co-purify with the histone deacetylases HDAC1 and HDAC2, as well as HP1, suggesting cooperative roles in chromatin modification and transcriptional repression [47].

**Paramyxovirus infection triggers stress-associated phosphorylation of TRIM28, but not loss of SUMO-modified TRIM28**

The finding that TRIM28 PB is a major target for V protein is intriguing given the importance of this domain in mediating transcriptional repression, and its harboring of important regulatory post-translational modification sites such as critical SUMO-modifications and the stress-regulated phosphorylation site at S824. SUMOylation of TRIM28 is critical for its role in gene silencing, facilitating the recruitment of SETDB1 and NuRD to establish repressive chromatin states [48,49]. In contrast, phosphorylation at S824 is linked to chromatin relaxation and is associated with transcriptional reactivation during hypoxia, heat-shock, or genotoxic stress responses [33,34,50,51]. Notably, S824 phosphorylation has been suggested to antagonize TRIM28 SUMOylation levels, potentially acting as a regulatory switch between its repressive and permissive states [33,52]. We therefore hypothesized that V protein engagement with the TRIM28 PB during infection may act to disrupt regulated TRIM28 functions. To this end, we sought to analyze differences in infection-triggered changes to TRIM28 between PIV2 (representing a virus encoding a V protein that engages efficiently with TRIM28) and PIV5 (representing a virus encoding a V protein that engages less efficiently with TRIM28). Furthermore, we aimed to compare these paramyxovirus infections to infection with IAV, given that IAV infection leads to loss of SUMO-modified TRIM28 and the consequent upregulation of ERE transcripts in infected cells [17]. Thus, we infected A549 cells with each virus at an MOI of 2-5 PFU/cell and subsequently prepared total cell lysates at various times post-infection. Western blot analysis revealed that all three (PIV2, PIV5, IAV) infections induced phosphorylation of TRIM28 at S824 (Fig 4A, B), suggesting that infection with each of these viruses results in a cellular stress response similar to that induced by the DNA damage-causing molecule etoposide, which served as a positive control for TRIM28 S824 phosphorylation. Interestingly, previous investigations of IAV-induced TRIM28 phosphorylation did not report upregulation of S824 phosphorylation upon IAV infection [17,53]. In this work, we used a different phospho-S824 TRIM28 antibody than in earlier reports and studied later times post-infection, which may potentially explain this discrepancy. Importantly, phosphorylation of TRIM28 at S824 has been reported to be associated with the loss of SUMO-modified TRIM28 [33,54], and we have previously demonstrated that IAV infection leads to loss of SUMO-modified TRIM28 [17]. S824 phosphorylation-dependent loss of SUMO-modified TRIM28 has been attributed to RNF4-mediated proteasomal degradation [54], though it may be that S824 phosphorylation instead interplays with SUMO protease-mediated deSUMOylation of TRIM28 [33].

To gain precise insights into the SUMOylation status of TRIM28 during infections with PIV2, PIV5 and IAV, we used a SUMO2/3-specific antibody to immunoprecipitate SUMO2/3-modified proteins from total cell lysates and assessed the SUMO2/3-modification status of TRIM28 by western blot. Despite the observed similarities in virus-induced TRIM28 S824 phosphorylation, the effects of infection on SUMO2/3-modified TRIM28 were markedly different between the viruses: while IAV infection led to a near-total loss of SUMO2/3-modified TRIM28, together with a global increase in total SUMO2/3 conjugates as previously described [17,55], neither PIV2 nor PIV5 infection resulted in any detectable loss of SUMO2/3-modified TRIM28 (Fig 4C, D). Indeed, quantification of results from independent replicates revealed a potentially subtle increase in SUMO2/3-modified TRIM28 levels in paramyxovirus-infected cells (Fig 4E, F). Interestingly, a previous study reported that SeV infection led to increased levels of SUMOylated TRIM28 in HEK293T cells

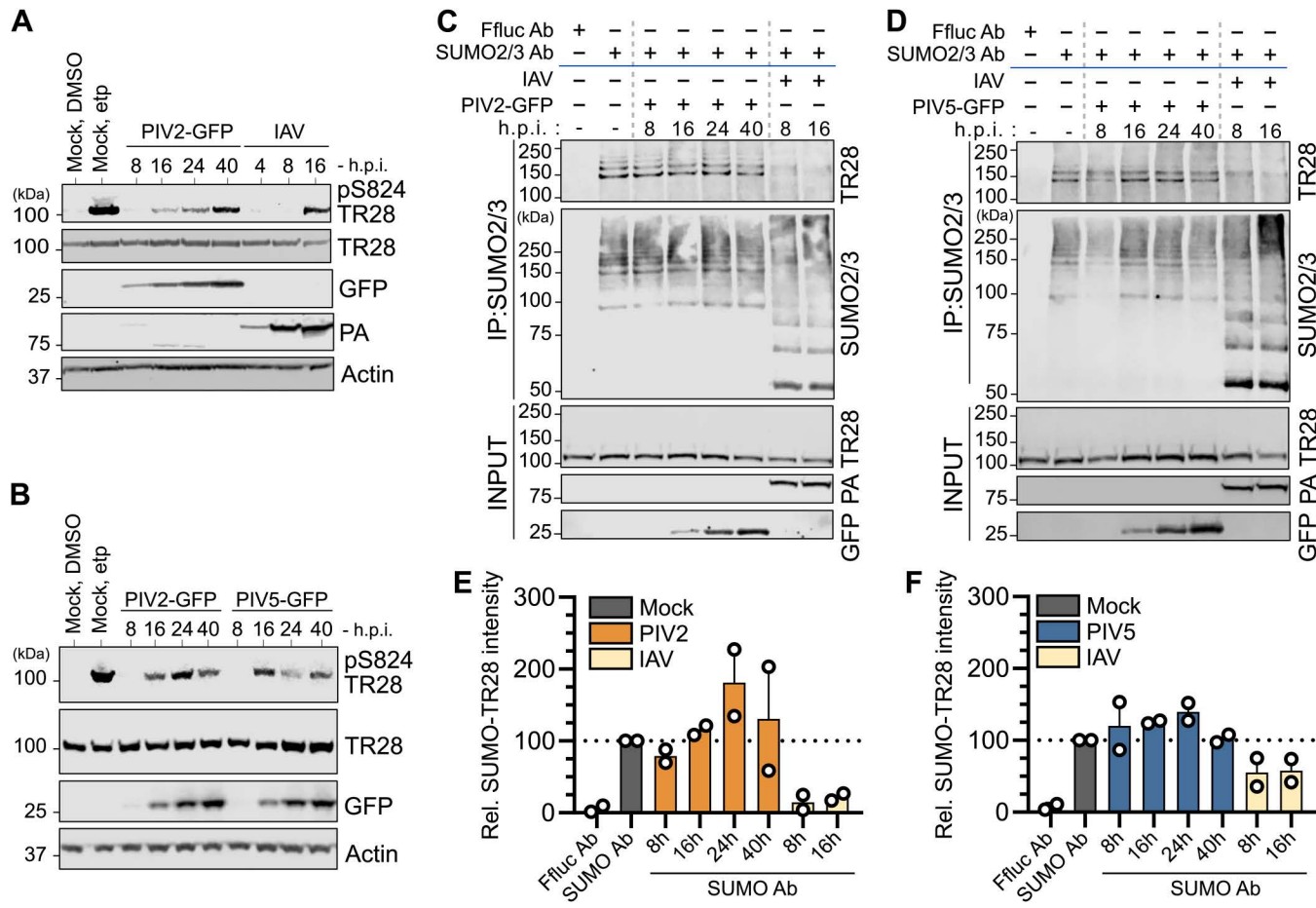

**Fig 4. Paramyxovirus infection triggers stress-associated phosphorylation of TRIM28, but not loss of SUMO-modified TRIM28.** (A) and (B) Analysis of TRIM28 phosphorylation at S824 during virus infections. A549 cells were infected (or mock) with PIV2-GFP or PIV5-GFP (MOI = 2 PFU/ cell), or IAV (MOI = 5 PFU/cell). Cell lysates were collected at the indicated times post-infection and western blot analysis was used to assess levels of the indicated proteins (phosphorylation of TRIM28 at S824: pS824 TR28). As a parallel positive control for TRIM28 phosphorylation at S824, cells were treated for 2 hours with 10 μM etoposide (etp). Data are representative of n = 3 independent experiments. (C) and (D) Analysis of SUMO-modified TRIM28 levels during PIV2-GFP (C) or PIV5-GFP (D) infections. A549 cells were infected similarly to experiments in (A), but cell lysates were subjected to immunoprecipitation (IP) using an anti-SUMO2/3 antibody or an anti-Ffluc antibody (control). The indicated co-precipitated proteins were analyzed by western blot. Input fractions were also analyzed. Data are representative of n = 2 independent experiments. (E) and (F) Quantification of SUMO-modified TRIM28 levels during virus infections. Band intensities of immunoprecipitated SUMO2/3-modified TRIM28 from experiments shown in (C) and (D) were normalized to input levels of TRIM28 and expressed relative (rel.) to the SUMO2/3-modified TRIM28 levels in the mock-infected condition. Data points correspond to results from individual experiments, and results from n = 2 independent experiments are shown. Bars represent mean values, with error bars representing range.

[56], supporting our observation that paramyxovirus infection does not reduce, and may even increase, SUMO2/3-modified TRIM28 levels. Overall, these data suggest a shared host response to paramyxovirus and IAV infections leading to stress-associated TRIM28 phosphorylation at S824. While this phosphorylation event has been linked to loss of SUMO-modified TRIM28 [33,54], which occurs during IAV infection, loss of SUMO-modified TRIM28 does not appear to occur during paramyxovirus infections. It is tempting to speculate that engagement of V with the PB domain of TRIM28, which occurs independently of TRIM28-S824 (Fig 3C), may act to limit loss of SUMO-modified TRIM28 during paramyxovirus infection.

**PIV2, but not PIV5, can dampen IAV-induced loss of SUMO-modified TRIM28 and limit derepression of a TRIM28-dependent ERE**

The precise molecular pathways by which some virus infections (particularly IAV) trigger loss of SUMO-modified TRIM28 and lead to derepression of EREs with potentiation of immune activation remain poorly understood [17,53,57]. This limits the experimental possibilities to dissect the potential for paramyxovirus V proteins alone to antagonize such responses. To overcome this, we therefore designed a strategy to assess whether pre-infection with a paramyxovirus encoding a V protein that engages efficiently with TRIM28 (PIV2) could limit subsequent IAV infection-induced loss of SUMO-modified TRIM28 and derepression of EREs. Thus, A549 cells were first infected (or mock) with PIV2 at an MOI of 5 PFU/cell for 24 hours, allowing V protein expression and its potential association with TRIM28. In parallel, A549 cells were similarly infected with PIV5 to represent a virus encoding a V protein that engages less efficiently with TRIM28. These infections were followed by a subsequent 10 hour super-infection with IAV (MOI of 5 PFU/cell) to allow for infection-triggered loss of SUMO-modified TRIM28. As shown by immunofluorescence microscopy, the desired infectivity was achieved for all viruses used, with approximately 90% of cells in co-infected conditions infected with PIV2 or PIV5, and approximately 80% with IAV (Fig 5A). Following on from this, we assessed the levels of SUMO2/3-modified TRIM28 in total cell lysates from each condition using SUMO2/3 immunoprecipitation followed by western blot. Remarkably, while IAV infection alone resulted in the clear loss of SUMO2/3-modified TRIM28, this loss was markedly reduced in cells pre-infected with PIV2 (Fig 5B). In contrast, pre-infection with PIV5 did not provide substantial protection against IAV-induced loss of SUMO2/3-modified TRIM28 (Fig 5B). Importantly, these differences were not due to differences in IAV infection/replication efficiency caused by the pre-infecting paramyxovirus, as determined by co-infection immunostaining (Fig 5A), assessment of IAV protein levels (Fig 5B), or assessment of changes in global SUMO2/3-conjugated protein levels (Fig 5B). The ability of PIV2, but not PIV5, pre-infection to limit IAV-induced loss of SUMO2/3-modified TRIM28 correlates well with our observation that PIV2-V, but not PIV5-V, efficiently engages with TRIM28. It may be that the more efficient engagement of PIV2-V with TRIM28 is necessary to suppress the strong heterologous IAV-triggered effects on TRIM28, while in the context of the homologous virus infections, the less efficient PIV5-V is sufficient to limit effects triggered by PIV5 (Fig 4D). Overall, these data suggest that V protein engagement with TRIM28 may limit virus-triggered host responses mediated by loss of SUMO-modified TRIM28.

We next sought to determine whether the efficient stabilization of SUMO-modified TRIM28 by PIV2 might influence the downstream derepression of EREs. To this end, we sequentially co-infected A549 cells as detailed above, but harvested cell lysates after 16 h of IAV infection to perform RT-qPCR analysis. Importantly, no significant differences were observed in viral transcript levels between single and co-infected conditions, indicating that co-infection did not impact the RNA levels of either paramyxovirus or IAV (Fig 5C, D, E), which is in line with the previous immunofluorescence and western blot data (Fig 5A, B). Consistent with prior observations [18], and the induced loss of SUMO-modified TRIM28, IAV infection alone led to a significant upregulation of several tested ERE RNAs, including HERVK14C, ERV9–1 and LTR13 (Fig 5F, G, H). In contrast, infections with PIV2 or PIV5 alone did not induce upregulation of these EREs to notable extents (Fig 5F, G, H). In the co-infection conditions, we observed that prior PIV2 infection significantly dampened the induction of HERVK14C and ERV9–1 RNAs by IAV infection, but exacerbated the effect of IAV-mediated induction of LTR13 (Fig 5F, G, H). Consistent with its minimal effect on IAV-triggered loss of SUMO-modified TRIM28, prior PIV5 infection did not limit IAV-triggered induction of EREs (Fig 5F, G, H). The results relating to HERVK14C are of particular interest, as this ERE is well-established to be strictly repressed by TRIM28 [58], making it a key benchmark ERE for assessing TRIM28 functionality. In this context, and similar to prior PIV2 infection, transfection of HEK293T cells with a plasmid expressing FLAG-tagged PIV2-V was sufficient to limit subsequent IAV-mediated induction of HERVK14C (Fig 5I, J). Thus, the ability of prior PIV2 infection, as well as expression of an efficient TRIM28-engaging V protein (PIV2-V) alone, to limit IAV-mediated induction of TRIM28-specific HERVK14C supports the hypothesis that a paramyxovirus V protein can limit stress-induced loss of SUMO-modified TRIM28 and thereby maintain transcriptional repression of certain EREs.

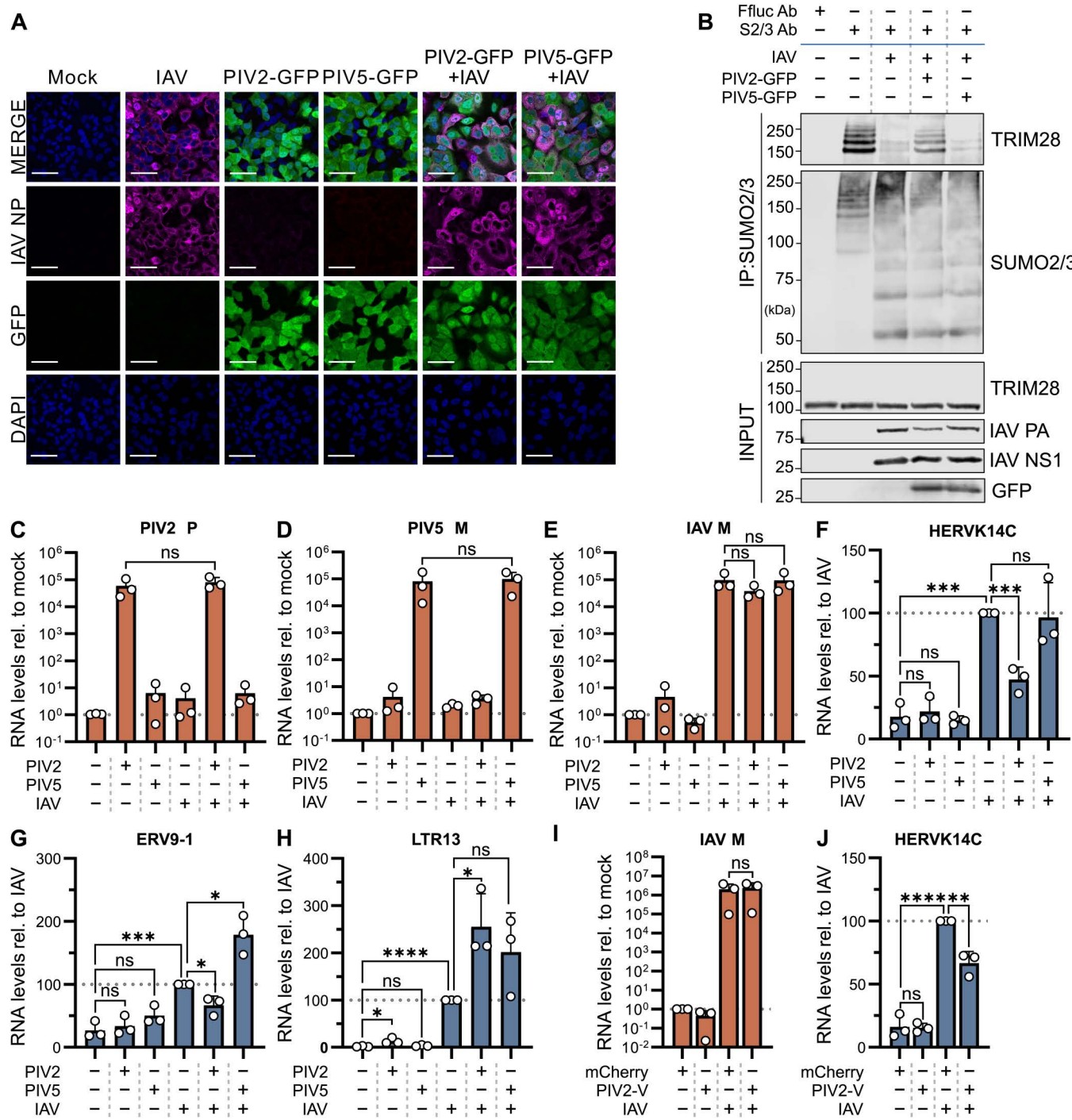

**Fig 5. PIV2, but not PIV5, can dampen IAV-induced loss of SUMO-modified TRIM28 and limit derepression of TRIM28-dependent EREs.** (A) Immunofluorescence microscopy of paramyxovirus and IAV single- or co- infections. A549 cells were infected (or mock) with PIV2-GFP or PIV5-GFP for 24 hours at MOI = 5 PFU/cell, before superinfection (or mock) with IAV for 10 hours at MOI = 5 PFU/cell. Cells were subsequently fixed, permeabilized, and stained with DAPI (nuclei, blue), anti-GFP antibody (paramyxovirus marker, green), and anti-NP antibody (IAV nucleoprotein, magenta). Images were acquired by confocal microscopy. Scale bars: 50 µm. Data are representative of n = 2 independent experiments. (B) Analysis of SUMO-modified TRIM28 levels during paramyxovirus and IAV co-infections. A549 cells were infected similarly to experiments in (A), but cell lysates were subjected to immunoprecipitation (IP) using an anti-SUMO2/3 antibody or an anti-Ffluc antibody (control). The indicated co-precipitated proteins were analyzed by western blot. Input fractions were also analyzed. Data are representative of n = 2 independent experiments. (C–H) RT-qPCR analysis of viral and host

ERE transcripts during paramyxovirus and IAV single- or co- infections. A549 cells were infected similarly to experiments in (A), but the IAV superinfection lasted 16 hours. RNA was extracted from cell lysates and subsequently analyzed by RT-qPCR. (C–E) Viral transcript levels (PIV2 P, PIV5 M, and IAV M) were measured relative to mock-infected cells. (F–H) Host ERE levels (HERVK14C, ERV9.1, and LTR13) were measured relative to the IAV only infected condition. (I-J) RT-qPCR analysis of viral and host ERE transcripts during IAV infection in the presence or absence of PIV2-V. HEK293T cells were transfected for 24 hours with FLAG-tagged constructs expressing mCherry or PIV2-V, followed by infection (or mock) with IAV for 16h at an estimated MOI of 2.5 PFU/cell. RNA was extracted from cell lysates and analyzed by RT-qPCR. (I) IAV M, and (J) HERVK14C, expression levels were measured relative to mock-infected or IAV-infected mCherry-transfected cells, respectively. For (C-J), relative quantities calculated using the ΔΔCt method normalized to 18S-rRNA. Data points correspond to results from individual experiments, and results from n = 3 independent experiments are shown. Bars represent mean values, with error bars representing standard deviation (SD). Appropriate comparisons were tested for statistical significance using an unpaired t test (*, P ≤ 0.05; **, P ≤ 0.01; ***, P ≤ 0.001; ****, P ≤ 0.0001; ns, non-significant).

## Discussion

Paramyxovirus V proteins are well-known antagonists of canonical innate immune signaling [29–32], yet our description of their engagement with TRIM28, a chromatin-associated transcriptional repressor, suggests a hitherto unknown additional activity. TRIM28, and its associated SUMO-modification, plays a key role in silencing transcription through nucleating chromatin remodeling complexes, and is particularly critical for the repression of young EREs [12–14]. Here, we demonstrate that multiple paramyxovirus V proteins can complex with TRIM28, both in transient overexpression- and infection- based co-immunoprecipitation assays. Given that V proteins have primarily been thought to target cytoplasmic immune pathways, such an interaction with a nuclear transcriptional repressor is notable, although a large body of existing data has indicated that V proteins can stably or transiently enter the nucleus [36–43]. Based on the data provided herein, TRIM28 engagement by V proteins may therefore provide insight into a previously unknown nuclear function for these viral virulence factors. Unlike IAV infection, PIV2 and PIV5 infections did not lead to loss of SUMO-modified TRIM28, despite all three infections inducing the stress-associated TRIM28 phosphorylation at S824 that has been linked to reduction in SUMO-modified TRIM28 levels [33,52,54]. Furthermore, pre-infection with PIV2 (or sole expression of its V protein that efficiently engages with TRIM28) limited subsequent IAV-triggered loss of SUMO-modified TRIM28 and/or derepression of selected EREs. Thus, paramyxovirus V proteins may have a function in engaging a nuclear TRIM28-containing complex to limit infection-triggered induction of certain EREs that may otherwise negatively impact virus replication via mechanisms including ERE-driven immune activation [6,10,15–17]. While the IAV NS1 protein may also limit ERE-driven immunity by sequestering ERE-derived dsRNA away from cytosolic sensors [18], the identification of paramyxovirus V proteins as potentially more direct inhibitors of this system lends further support to the concept that a TRIM28-regulated ERE-based antiviral defense is a *bona fide* pathway to which some viruses have evolved antagonistic strategies.

The precise triggers and consequent mechanisms by which some viral infections lead to loss of SUMO-modified TRIM28 remain largely unclear, which currently limits detailed molecular studies during paramyxovirus infections. However, in different contexts, several pathways for this loss have been identified, including TRIM28 deSUMOylation by SUMO-specific proteases, phosphorylation-dependent recruitment of SUMO-targeted ubiquitin ligases (such as RNF4) for ubiquitination-dependent degradation of SUMO-modified TRIM28, and destabilization of SUMO-modified TRIM28 via XAF1 during immune activation [17,33,53,54,57,59]. Given that a major TRIM28 target domain for V proteins appears to be the PB domain that harbors the lysine residues primarily modified by SUMO paralogs, it will be important to determine which of these processes may be influenced by V protein engagement with TRIM28. The exact molecular mechanisms underlying the interplay of V with TRIM28 may be complex, given our observation that V proteins also appear to engage with the CC domain of TRIM28. In addition, it is worth noting that we found evidence that V proteins can also potentially engage with the PB domains of other TIF-1 family members, such as TRIM24, TRIM33, and TRIM66. The exact functional coordination of these interactions, whether single or multiple complexes form, and which factor is the direct target for V, remain open questions to dissect in future studies. TIF-1 family members have been found together in the same complexes previously, and can exhibit both functional overlaps with TRIM28 and independent activities [46,47]. For example,

similar to TRIM28, TRIM24 undergoes ATM-mediated phosphorylation during the DNA damage response, but this leads to TRIM24 destabilization and upregulation of pro-apoptotic p53, a defined ubiquitination target for TRIM24 [60,61]. Nevertheless, knock-out of TRIM24 can lead to ERE derepression and triggering of an IFN response, drawing clear functional parallels with TRIM28 [62]. TRIM33 and TRIM66 are also implicated in DNA damage responses [46,63,64], acting co-operatively with TIF-1 family members for transcriptional silencing [62]. Similar to TRIM28, TRIM33 has also been implicated in additional activities, such as mediating ubiquitination events that can activate NLRP3 inflammasomes [65,66]. It will be worthwhile to dissect the potential consequences of V protein engagement with different TIF-1 family members and how this might contribute to immune or inflammatory evasion during paramyxovirus infection.

A key striking finding of our study was the observation that V proteins from distinct paramyxoviruses appear to engage differentially with human TRIM28. While we identified a single amino acid position in the CTDs of PIV2 and PIV5 V proteins that clearly influences this phenotype, candidate contributors to the phenotypes of the other paramyxovirus V proteins tested were not obvious from primary sequence alignments. This suggests that the sequence determinants in V responsible for such differential engagement are highly complex. Notably, the potential cellular localization of V proteins from different paramyxoviruses may correlate well with the variation in their abilities to co-immunoprecipitate TRIM28. For instance, PIV2-V, which was reported to be predominantly nuclear [43,67], efficiently precipitates TRIM28, whereas PIV5-V, which might exhibit a more diffuse nuclear-cytoplasmic distribution [40,42], shows less efficient TRIM28 precipitation capacity. In this regard, the region of PIV2-V that we identified as important for engagement with TRIM28 has previously been implicated in its strong nuclear targeting and retention [36]. Furthermore, NiV-V, which is mostly cytoplasmic but can transiently shuttle into the nucleus [37,68], was identified by us as being one of the weakest engagers of human TRIM28. Extensive further mapping efforts are therefore required to dissect TRIM28 precipitation differences between paramyxovirus V proteins. Such studies will be essential before any functional correlations of TRIM28 engagement can be made with respect to potential paramyxovirus host-range determinants or viral virulence. Nevertheless, this will be an intriguing area of future study.

Taken together, our work provides evidence suggesting a previously unrecognized function of paramyxovirus V proteins in engaging and modulating the action of host nuclear TRIM28. While V proteins are well-established as antagonists of canonical cytoplasmic IFN signaling pathways, our new findings raise the possibility that V proteins also engage TRIM28 to suppress infection-triggered ERE-driven antiviral mechanisms. Our data therefore add to the emerging evidence that virus infections trigger, and therefore must at least partially antagonize, ERE-based antiviral defense systems.

## Methods

### Cells, compounds and viruses.

A549, MDCK, Vero CCL-81 and HEK293T cells were cultured in Dulbecco's Modified Eagle's Medium (DMEM) (Life Technologies) supplemented with 10% (vol/vol) fetal bovine serum (FBS), 100 units/mL penicillin, and 100 µg/mL streptomycin (Gibco Life Technologies). TRIM28-knockout HEK293T cells (HEK293T-T28KO) [58] were a kind gift from Helen Rowe (Queen Mary University of London, UK). Where indicated, etoposide (E1383, Sigma-Aldrich) was added to the culture medium at the indicated concentration. IAV strain A/WSN/1933 (H1N1) was propagated in MDCK cells. PIV5 (strain W3A), PIV2-GFP (P222, ViraTree; strain V94) and PIV5-GFP (P523, ViraTree; strain W3A) were propagated in Vero CCL-81 cells. PIV5-VΔC [35] was propagated in Vero-V1 cells (kindly provided by Steve Goodbourn, St. George's, University of London, UK). Virus stocks were titrated using standard plaque assays for IAV, PIV5-W3A, and PIV5-VΔC, and focus forming assays for GFP-expressing viruses.

### Virus infections.

Generally, cells were seeded at $6 \times 10^5$ cells per well of a 6-well plate or $1 \times 10^5$ cells per well of a 24-well plate and infected the next day at the indicated multiplicity of infection (MOI). For infections with PIV2 and PIV5, inoculum was

prepared in DMEM supplemented with 2% FBS, 100 units/mL penicillin, and 100 µg/mL streptomycin. Cells were then incubated with the inoculum for 1 h at 37°C, washed with PBS, and then overlaid with DMEM supplemented with 2% FBS, 100 units/mL penicillin, and 100 µg/mL streptomycin. For infections with IAV, inoculum was prepared in PBS supplemented with 0.3% BSA, 1 mM $Ca^{2+}/Mg^{2+}$, 100 units/mL penicillin, and 100 µg/mL streptomycin. Cells were then incubated with the inoculum for 1 h at 37°C, washed with PBS, then overlaid with DMEM supplemented with 0.1% FBS, 100 units/mL penicillin, 100 µg/mL streptomycin, 0.3% BSA, 20 mM HEPES, and 1 µg/mL TPCK-trypsin (Sigma-Aldrich). TPCK-trypsin was omitted from the overlay medium when IAV infections of HEK293T cells were performed.

**Plasmids.**

The pCMV7.1_3xFlag-mCherry [55] and pEF-FLAG-V [30,69,70] expression vectors were previously described. Sequences corresponding to N- and C-terminal domains of MeV-V (1–216, 207–299), MuV-V (1–152, 149–224), and PIV5-V (1–161, 157–222) were PCR-amplified from the corresponding pEF-FLAG-V plasmids and cloned back into pEF-FLAG via NcoI and XbaI. Full-length PIV2-V and PIV5-V were PCR-amplified from pre-existing vectors and cloned into pCMV7.1_3xFlag via NotI and EcoRI. The pCS2-HA-TRIM28ΔPB plasmid (Addgene #124959 [71], a kind gift from Michelle Barton, University of Texas MD Anderson Cancer Center, Texas, USA) was used as a source of the pCS2-HA expression vector. Full-length human TRIM28 was subcloned from a pre-existing plasmid [17] into the pCS2-HA vector via FseI and XbaI sites. TRIM28 truncation mutants were generated by PCR amplification of defined subdomains and cloned into the pCS2-HA vector through the same restriction sites. Chimeric PIV2-V and PIV5-V protein-encoding cDNAs were generated by overlap-extension PCR in the pCMV7.1_3xFlag backbone, and point mutations were generated using the QuikChange II XL Site-Directed Mutagenesis Kit (200521, Agilent Technologies). The pcDNA3.1 expression vectors containing the coding sequences of the HA-tagged PB domains from TRIM24, TRIM33, and TRIM66 were synthesized using GeneArt (Thermo Fisher Scientific). Generally, HEK293T cells at ~70% confluency were transfected with the described plasmids using FuGENE HD (E5911, Promega), following the manufacturer's instructions.

**RNA extraction and RT-qPCR.**

Cells were processed with the ReliaPrep RNA Cell Miniprep kit (Promega) according to the manufacturer's protocol. 1 µg of RNA was reverse-transcribed into cDNA using SuperScript IV First-Strand Synthesis System (18091300, Thermo Fisher Scientific) with random primers (Promega). qPCR was performed in technical duplicates with the PowerTrack SYBR Green Master Mix (A46109, Thermo Fisher Scientific) in an ABI7300 Real-Time PCR system (Applied Biosystems). qPCR primers used for specific targets are listed in S1 Table. Data were processed using SDS Shell, and relative cDNA quantities were calculated according to the ΔΔCt method in relation to 18S-rRNA.

**Immunoprecipitation assays.**

A549 or HEK293T cells were washed once with PBS, and subsequently lysed using co-IP lysis buffer (50 mM Tris-HCl (pH 7.4), 300 mM NaCl, 1 mM EDTA, 1% Triton X-100, 10 mM NEM) supplemented with cOmplete Protease Inhibitor Cocktail (11836170001, Roche) and PhosSTOP (PHOSS-RO, Roche). Lysates were sonicated on ice to shear nucleic acids, cleared by centrifugation at 16,000 x g for 15 minutes at 4°C, then subjected to overnight incubation with 2 µg of primary antibody at 4°C with rotation. Antibodies used were: mouse anti-V5 monoclonal antibody (MCA1360, BioRad), rabbit anti-TRIM28 polyclonal antibody (A-300-274A, Bethyl), and rabbit anti-firefly luciferase (Ffluc) polyclonal antibody (ab21176, abcam). Immunocomplexes were then captured with Protein G Sepharose 4 Fast Flow resin (17061801, Cytiva) for 2 hours at 4°C with rotation. For FLAG immunoprecipitations, lysates were directly incubated with anti-FLAG M2 Affinity Gel (A2220, Millipore) for at least 2 hours at 4°C with rotation. Beads were subsequently washed five times with lysis buffer or TBS (FLAG affinity gel), and bound proteins were eluted with urea disruption buffer (6 M urea, 2 M βME, 4% SDS, bromophenol blue).

For SUMO-IPs, A549 cells (1.2 x 10$^6$ cells per 6-well) were infected for the indicated times with IAV (MOI = 5 PFU/cell), PIV2-GFP, or PIV5-GFP (both MOI = 2 PFU/cell). At the time of harvest, cells were washed once with PBS, and total lysates were collected using SUMO lysis buffer (50mM Tris-HCl (pH 7.8), 650mM NaCl, 1% NP-40, 2% SDS, 5mM EDTA, 20mM NEM, 10mM βME) supplemented with cOmplete Protease Inhibitor Cocktail. Lysates were then diluted 1:1 in modified wash buffer (50mM Tris-HCl (pH 7.8), 1% NP-40, 0.1% SDS, 5mM EDTA) supplemented with cOmplete Protease Inhibitor Cocktail and sonicated on ice to shear nucleic acids. Lysates were cleared by centrifugation at 16,000 x g for 30 minutes at 4°C, then subjected to overnight incubation with 2 µg of rabbit anti-SUMO2/3 polyclonal antibody (ab3742, abcam) or rabbit anti-firefly luciferase (Ffluc) polyclonal antibody (ab21176, abcam). Magnetic protein G Dynabeads (10004D, Thermo Fisher) were added and incubated for at least 30 minutes at room temperature with rotation. Subsequently, beads were washed five times with modified wash buffer, and bound proteins were eluted with urea disruption buffer.

**Western blot analyses.**

Cells or protein fractions were lysed in urea disruption buffer, and nucleic acids were sheared by sonication. Following treatment at 95°C for 5 min, proteins were separated by SDS-PAGE using Bolt (Invitrogen) Bis-Tris Plus Mini Protein Gels (4–12%) and transferred onto 0.45 µm nitrocellulose membranes (Amersham) using the Invitrogen Mini Gel Tank and Blot Module Set. Membranes were blocked in 5% milk-TBS-T (5% milk in TBS supplemented with 0.1% Tween 20), or 5% BSA-TBS-T for phosphorylated protein detection, and incubated with the following primary antibodies: rabbit anti-TRIM28 polyclonal antibody (A-300-274A, Bethyl); rabbit anti-phospho-TRIM28 (S824) polyclonal antibody (A-300-767A, Bethyl); mouse anti-FLAG M2 monoclonal antibody (F1804, Sigma Aldrich); rabbit anti-FLAG polyclonal antibody (F7425, Sigma Aldrich); mouse anti-V5 monoclonal antibody (MCA1360, BioRad); mouse anti-DDB1 monoclonal antibody (2B12D1, Invitrogen); rabbit anti-HA monoclonal antibody (C29F4, Cell Signaling Technologies); rabbit anti-GFP polyclonal antibody (GTX113617); rabbit anti-PA polyclonal antibody (GTX118991, Genetex); rabbit anti-NS1 polyclonal antibody (GTX125990, Genetex); rabbit anti-SUMO2/3 polyclonal antibody (ab3742, abcam); or mouse anti-actin monoclonal antibody (sc-47778, Santa-Cruz). After washing with TBS-T, membranes were incubated in 5% milk in TBS-T with the following secondary antibodies: IRDye 800CW goat anti-mouse IgG (926–32210, LI-COR Biosciences), IRDye 800CW goat anti-rabbit IgG (926–32211, LI-COR Biosciences), IRDye 680RD goat anti-mouse IgG (926–68070, LI-COR Biosciences), or IRDye 680RD goat anti-rabbit IgG (926–68071, LI-COR Biosciences). Membranes were imaged with the Odyssey Fc Imager (LI-COR Biosciences) and quantified with the Image Studio Lite Quantification software (LI-COR Biosciences).

**Immunofluorescence microscopy.**

Cells seeded onto glass coverslips (poly-l-lysine (P4832, Sigma-Aldrich) coated for HEK293Ts) were treated as indicated prior to washing with PBS, fixing with 3.5% PFA for 15 minutes at room temperature, and permeabilizing for at least 1 hour with confocal buffer (PBS supplemented with 50 mM ammonium chloride, 0.1% saponin, and 2% BSA). Primary and secondary antibodies were diluted in confocal buffer and incubated with the samples for at least 1 hour. The following primary antibodies were used: rabbit anti-GFP polyclonal antibody (GTX113617, GeneTex), mouse anti-NP monoclonal antibody (derived from H16-L10-4R5, ATCC), rabbit anti-TRIM28 polyclonal antibody (A-300-274A, Bethyl), and mouse anti-FLAG M2 monoclonal antibody (F1804, Sigma Aldrich). Nuclei were stained with DAPI (10236276001, Sigma-Aldrich). Coverslips were mounted using ProLong Gold Antifade Mountant (P36930; Thermo Fisher Scientific), and images were captured using a Leica SP8 confocal microscope. Further processing of images was done using ImageJ/Fiji programs.

**Data analysis.**

Statistical analyses were generally performed using GraphPad Prism 7. The specific statistical tests and corresponding P values are indicated in the figure legends. RT-qPCR data were analyzed using the $2^{-\Delta\Delta CT}$ method as described previously [72]. Protein sequence alignments were performed with the Clustal Omega program using MegAlign Pro.

## Supporting information

**S1 Fig. Protein sequence alignments of TRIM28 PHD-Bromodomains and Coiled-Coil domains across species.** (A-B) Protein sequence alignments of Coiled-Coil domains (A) and PHD-Bromodomains (B) from TRIM28 orthologs from the indicated potential paramyxovirus host species, as determined by Clustal Omega. Numbers reference amino-acid position in the specific domain. Red coloring indicates a difference as compared with the *Homo sapiens* sequence. (EPS)

**S1 Table. Primer pairs used for RT-qPCR analyses.** (DOCX)

**S1 Dataset. Quantitative source data underlying panels in Figs 1, 2, 4, and 5.** (XLSX)

## Acknowledgments

We thank Steve Goodbourn (St George's, University of London, UK) for advice and for the generous sharing of essential reagents. We are also grateful to Helen Rowe (Queen Mary University of London, UK) and Michelle Barton (University of Texas, USA) for providing important tools.

## Author contributions

**Conceptualization:** Gauthier Lieber, Nora Schmidt, Benjamin G. Hale.

**Data curation:** Gauthier Lieber, Florence Kwaschik.

**Formal analysis:** Gauthier Lieber, Florence Kwaschik, Nora Schmidt.

**Funding acquisition:** Benjamin G. Hale.

**Investigation:** Gauthier Lieber, Florence Kwaschik.

**Methodology:** Gauthier Lieber, Florence Kwaschik, Marie Lork, Nora Schmidt.

**Project administration:** Benjamin G. Hale.

**Resources:** Marie Lork, Benjamin G. Hale.

**Supervision:** Marie Lork, Benjamin G. Hale.

**Validation:** Gauthier Lieber, Florence Kwaschik.

**Visualization:** Gauthier Lieber, Florence Kwaschik.

**Writing – original draft:** Gauthier Lieber, Marie Lork, Nora Schmidt, Benjamin G. Hale.

**Writing – review & editing:** Gauthier Lieber, Florence Kwaschik, Marie Lork, Nora Schmidt, Benjamin G. Hale.

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
