## [Decision Letter · Decision Letter 0]

22 Jun 2025

TRIM28 is a Target for Paramyxovirus V Proteins

PLOS Pathogens

Dear Dr. Hale,

Thank you for submitting your manuscript to PLOS Pathogens. After careful consideration, we feel that it has merit but does not fully meet PLOS Pathogens's publication criteria as it currently stands. Therefore, we invite you to submit a revised version of the manuscript that addresses the points raised during the review process.

Please submit your revised manuscript within 60 days Aug 21 2025 11:59PM. If you will need more time than this to complete your revisions, please reply to this message or contact the journal office at plospathogens@plos.org. Please include the following items when submitting your revised manuscript:

We look forward to receiving your revised manuscript.

Kind regards,

Jacob S. Yount

Academic Editor

PLOS Pathogens

Thomas Hoenen

Section Editor

PLOS Pathogens

Editor-in-Chief

PLOS Pathogens

orcid.org/0000-0003-2946-9497

Editor-in-Chief

PLOS Pathogens

orcid.org/0000-0002-7699-2064

**Additional Editor Comments :**

Most of the comments can likely be addressed with text responses and minor changes, though I would suggest experimentally addressing the following critiques:

1. Reviewer 1 requests evidence that V/TRIM28 interaction is not a post-lysis artifact given the disparate localizations of the two proteins.

2. Co-infection experiments requested by Reviewer 1. IAV infection of V-transfected cells may be the most straightforward experiment.

**Journal Requirements:**

At this stage, the following Authors/Authors require contributions: Gauthier Lieber, Marie Lork, Nora Schmidt, and Benjamin G. Hale. Please ensure that the full contributions of each author are acknowledged in the "Add/Edit/Remove Authors" section of our submission form.

- TM on pages: 15, and 16.

5) We note that your Data Availability Statement is currently as follows: "All relevant data are within the manuscript and its Supporting Information files.". Please confirm at this time whether or not your submission contains all raw data required to replicate the results of your study. Authors must share the “minimal data set” for their submission. PLOS defines the minimal data set to consist of the data required to replicate all study findings reported in the article, as well as related metadata and methods (https://journals.plos.org/plosone/s/data-availability#loc-minimal-data-set-definition).

**Reviewers' Comments:**

Reviewer's Responses to Questions

**Part I - Summary**

Reviewer #1: The authors show a novel interaction between TRIM28 and the paramyxovirus V protein. Using coIPs, the authors show interaction between TRIM28 and the V proteins of several paramyxoviruses. Furthermore, the authors map the interaction between PIV2 and PIV5 V proteins and TRIM28 to the C-terminus of V and the domains of TRIM28 interacting with V. Point mutations are then generated to test the importance of specific residues necessary for TRIM28-V interactions. A few issues need to be addressed.

Reviewer #2: The manuscript by Lieber et al. characterizes a novel TRIM28 binding function by the V protein of different paramyxoviruses. TRIM28, when SUMOylated can repress transcription of endogenous retroelements, which can have an impact on antiviral immunity, as in the case of influenza A virus, as described by the authors. Here, they described the TRIM28-V protein interaction and identify key regions of the V protein and TRIM28 where the interaction occurs. Interestingly there are differences in TRIM28 affinity depending on which paramyxovirus V protein is tested, the implications of which are not completely understood and can be further studied. Also of note was that this interaction did not lead to loss of ERE repression that is seen during influenza infection. Pre-infection with PIV-2, whose V protein has a strong interaction with TRIM28, prevented the loss of TRIM28 SUMOylation seen during subsequent influenza infection, significantly impacting ERE transcription.

The authors present their story in a compelling fashion and in a logical step-wise manner that is easy to follow and coherent. They convincingly show that V-TRIM28 interaction occurs and through specific C terminal domains in the V protein interacting with CC and PHB domains of TRIM28. The authors bring up several current unknowns and possible areas for future study in the discussion. I have only minor comments and questions for the authors

For the initial experiments, the authors perform coIP at 24 hours post-transfection and post-infection. During infection, what are the kinetics of V protein expression? Would it be of interest to examine how early V is expressed and begins to interact with TRIM28? Conversely, particularly during infection, which may last days, how stable is the V-TRIM28 interaction and for how long can EREs remain repressed? Presumably new V is produced during infection of new cells and continues this effect. The authors use a high MOI to produce infection in a high % of cells. But if a lower MOI is used and a small fraction of cells are infected initially, does the stress response in infection cells lead to derepression of EREs and possible innate immune stimulation in neighboring, uninfected cells that may negate the impact of this effect in infected cells?

In fig 1 the anticipated sizes of the V proteins of various paramyxoviruses varies considerably with HeV, NiV, and SeV are approximately 2x as large as others. Is this expected? This is my naivety with respect to different viruses in the family.

For figure 4, are C and D simply replicates of the same experiments? There are no differences between the panels other than the blot bands, but it is not explicitly stated anywhere that this is the case, so it was a bit confusing. The same for Fig 3 C and D. these are different domains being tested so that can be gleaned from looking at the labels, but it is not obvious and not obvious from the figure caption either.

For figure 3, the experiments were done in TRIM28 KO cells but for the experiments with TRIM24, 33, 66 were these also knocked out? Since you’re only looking at the specific domains, it is less of a concern that what you’re seeing in the blot is FL protein as a false positive, however this may impact V-TRIM interactions.

In the methods for influenza infection, the authors do not indicate if they used exogenous trypsin to aid in infection. They use a high MOI and look at the effects shortly after infection, so it is probably not an issue because they don’t need efficient replication of the virus. However a statement about this could be added.

The authors interestingly note that V proteins are usually cytoplasmic, and this interaction with nuclear proteins is novel. Is anything known about nuclear localization of V proteins? The authors mention previous studies showing it in the nucleus but with no known function. Is the C terminal domain critical also for nuclear localization? The authors could look at how it locates and important domains for this. They touch on this and speculate but did not explore it experimentally.

The authors do not actually look at any innate immune signaling that may or may not be triggered by transcription of EREs in their transfection or infection systems. This would be an important downstream effect that could shed light on outcomes of infection, at least in vitro.

Is there a reason why there are no

To me the most interesting finding is regarding NiV V and how it’s TRIM28 precipitation was less efficient, though it would probably be thought of as the most virulent of these viruses in humans. This is raised by the authors in the discussion however it seems to me that if this is the case, then during infection of hosts (particularly humans or other susceptible animals) that even if V is suppressing ERE transcription and this does have an impact on innate immunity, the effect likely is not impacting infection outcomes in the host and plays only a very minor role. It also could be that other innate immune signaling via other means dwarfs that being induced by EREs. Something interested to pursue further.

**Part II – Major Issues: Key Experiments Required for Acceptance**

Reviewer #1: Figure 1. (applies for all CoIPs). TRIM28 is thought to be localized to the nucleus and V protein is localized to the cytoplasm. It would be important to isolate nuclear and cytoplasmic fractions separately and not total cell lysates to show that the interaction observed between TRIM28 and V is not an artifact of the lysis procedure.

Figure 2.

- Panel 1A shows MeV V migrating close to the 37kDa marker. However, in Figure 1, MeV V migrates close to the 25kDa marker. Please explain and clarify if this is a mistake.

- Panel 2B - mCherry migrates at a different position compared to Figure 1A. The migration pattern between mCherry, PIV2, and PIV5 is similar, but the marker is not. Please clarify if figure 1 is correct.

- Panel 2E, G, I - Plotting relative co-IP on different axis types (log vs linear) prevents the reader from easily comparing results between panels. Please use a uniform axis type.

Figure 3 - Since the similarity between TRIM28 across species is highly conserved, it would be useful to include locations of non-conserved residues between species as part of the schematic. According to the sequence identify, there should only be 5-15 changes between humans and mice. would be easiest to show as a sequence alignment in supplement.

Figure 3 Panel B,C, and E - MeV V is above 37kDa. However, in Figure 1a, MeV V is close to 25kDa. Explain.

Figure 4 - Panel E and F - How can you make error bars representing Standard Deviation using an N of 2? Please change to range.

Co-infection studies

In addition to coinfections, it would be very important to show the effects of PIV2/PIV5 coinfection with influenza are due to the V protein or a different viral protein. Transfection of cells with V protein expression plasmids, followed by infection with influenza would greatly strengthen the arguments proposed by the author if the results are the same. Particularly when it comes to SUMO2/3 activity and ERE activation/RNA levels. One more thing, the authors have a delta V PIV5 virus. It would be highly relevant to include this virus with the coinfection studies to show the effects of delta V PIV5 on both SUMO2/3 activity and ERE activation.

Reviewer #2: (No Response)

**Part III – Minor Issues: Editorial and Data Presentation Modifications**

Reviewer #1: Figure 5) Fluorescence microscopy showing the localization of PIV2/PIV5 V and TRIM28 would greatly strengthen the manuscript.

Reviewer #2: (No Response)

PLOS authors have the option to publish the peer review history of their article (what does this mean? ). If published, this will include your full peer review and any attached files.

**Do you want your identity to be public for this peer review?** For information about this choice, including consent withdrawal, please see our Privacy Policy .

Reviewer #1: No

Reviewer #2: No

**Figure resubmission:**

**Reproducibility:**



---

## [Editor Report · Decision Letter 1]

26 Aug 2025

Dear Prof Hale,

We are pleased to inform you that your manuscript 'TRIM28 is a target for paramyxovirus V proteins' has been provisionally accepted for publication in PLOS Pathogens.

Best regards,

Thomas Hoenen

Section Editor

PLOS Pathogens

Thomas Hoenen

Section Editor

PLOS Pathogens

Sumita Bhaduri-McIntosh

Editor-in-Chief

PLOS Pathogens

orcid.org/0000-0003-2946-9497

Michael Malim

Editor-in-Chief

PLOS Pathogens

orcid.org/0000-0002-7699-2064

The authors have sufficiently addressed the reviewers' comments.
---

## [Editor Report · Acceptance letter]

Dear Prof Hale,

We are delighted to inform you that your manuscript, "TRIM28 is a target for paramyxovirus V proteins," has been formally accepted for publication in PLOS Pathogens.

Best regards,

Sumita Bhaduri-McIntosh

Editor-in-Chief

PLOS Pathogens

orcid.org/0000-0003-2946-9497

Michael Malim

Editor-in-Chief

PLOS Pathogens

orcid.org/0000-0002-7699-2064